# Diagnostic Implications of NGS-Based Molecular Profiling in Mature B-Cell Lymphomas with Potential Bone Marrow Involvement

**DOI:** 10.3390/diagnostics15060727

**Published:** 2025-03-14

**Authors:** Bernhard Strasser, Sebastian Mustafa, Josef Seier, Erich Wimmer, Josef Tomasits

**Affiliations:** 1Institute of Clinical Chemistry and Laboratory Medicine, Klinikum Wels-Grieskirchen 1, 4600 Wels, Austria; sebastian.mustafa@klinikum-wegr.at (S.M.); josef.seier@klinikum-wegr.at (J.S.); erich.wimmer@klinikum-wegr.at (E.W.); 2Department of Internal Medicine, Johannes Kepler University, 4040 Linz, Austria; josef.tomasits@kepleruniklinikum.at

**Keywords:** B-cell lymphoma, next-generation-sequencing, NGS, molecular diagnostics

## Abstract

**Background:** Methods such as cytogenetics and immunocytology/immunohistology provide essential diagnostic insights but may be limited in ambiguous cases of mature B-cell lymphoma. Next-generation sequencing (NGS) has emerged as a potential tool to improve diagnostics. **Methods:** We validated the analytical performance of a lymphoid customized NGS panel. Clinical validation was conducted in 226 patients with suspected mature B-cell lymphoma with potential bone marrow involvement across multiple clinically relevant scenarios. **Results:** NGS (1) achieved 100% sensitivity and specificity with high reproducibility (r = 0.995), confirming its analytical performance. (2) It reliably detected WHO-classified markers, including *BRAF* mutations in all hairy cell leukemia cases, *MYD88/CXCR4* mutations in lymphoplasmacytic lymphoma, and absence of *BRAF* mutations in splenic B-cell lymphoma with prominent nucleoli. (3) In lymphoma exclusion diagnostics, NGS identified mutations in previously undiagnosed cases, including a *BCORL1* mutation leading to reclassification as marginal zone lymphoma. (4) Among 105 confirmed lymphomas, 65% harbored mutations, with detection rates highest in HCL and LPL (100%) and CLL (62%), while follicular lymphoma showed no detectable mutations. (5) In cases with non-interpretable cytogenetics, NGS detected pathogenic variants in 61% of patients, compensating for inconclusive findings. (6) In cases with limited morphological assessment, NGS identified relevant mutations in 70%, outperforming cytogenetics (30%; *p* = 0.0256, OR = 5.44). **Conclusions:** NGS enhances the diagnostic accuracy of mature B-cell lymphomas by complementing traditional methods, refining WHO-classified subtypes, and improving detection in cases with inconclusive cytogenetics or morphology. NGS may reduce the need for unnecessary bone marrow re-punctures by providing additional information in ambiguous cases.

## 1. Introduction

The World Health Organization (WHO) 5th Edition of the Classification of Haematolymphoid Tumours (WHO-HAEM5) introduced a refined framework for the categorization of mature B-cell neoplasms, integrating molecular and genetic data alongside traditional morphological and immunophenotypic assessments. This latest classification emphasizes the role of molecular diagnostics in hematopathology, offering structured “essential” and “desirable” criteria to facilitate standardized lymphoma classification worldwide. A major development in this edition is the subdivision of mature B-cell neoplasms into 12 distinct families, incorporating new entities and refining disease definitions based on genomic and transcriptomic insights [1].

Despite these advances, the diagnosis of low-grade B-cell lymphoproliferative disorders remains a challenge due to overlapping clinical, morphological, and immunophenotypic features. Traditional approaches, including morphology, immunohistology [2,3], and flow cytometry [4,5], provide essential diagnostic information. However, the increasing availability of next-generation sequencing (NGS) and molecular profiling has raised new possibilities for refining disease classification, risk assessment, and therapeutic stratification [6,7].

For decades, cytogenetics has been the cornerstone of lymphoma classification. Conventional karyotyping, fluorescence in situ hybridization (FISH), and array comparative genomic hybridization are widely used to detect hallmark chromosomal rearrangements that define major lymphoma entities. These include t(14;18) in follicular lymphoma (FL), t(11;14) in mantle cell lymphoma (MCL), and t(8;14) in Burkitt lymphoma (BL). Furthermore, cytogenetic profiling is crucial in identifying *MYC*, *BCL2*, and *BCL6* rearrangements in high-grade B-cell lymphomas (HGBL-*MYC/BCL2/BCL6*), which define cases with aggressive clinical behavior and poor prognosis [8]. Alongside cytogenetics, FCM remains a frontline diagnostic tool, offering rapid and high-throughput characterization of immunophenotypic profiles. However, while FCM and cytogenetics are highly effective for initial diagnosis, they may lack the ability to resolve ambiguous cases where molecular profiling could provide additional information [9,10].

The integration of molecular genetics has already changed the prognostication of mature B-cell neoplasms. Recent genomic studies have identified key mutations that stratify patients based on their prognosis, helping to distinguish indolent from aggressive disease courses and guiding therapeutic decision making. Several molecular alterations have emerged as critical prognostic markers in B-cell neoplasms, reflecting their impact on disease evolution and treatment resistance. Among them, *TP53* mutations, frequently found in chronic lymphocytic leukemia (CLL), mantle cell lymphoma (MCL), and diffuse large B-cell lymphoma (DLBCL), are associated with chemo-refractoriness, rapid progression, and significantly reduced survival. Similarly, mutations in the *NOTCH1* pathway correlate with shorter overall survival and increased risk of histologic transformation, particularly in CLL, MCL, and splenic marginal zone lymphoma (SMZL). In follicular lymphoma (FL) and germinal center B-cell-like DLBCL, alterations in *EZH2* are linked to high-grade transformation, whereas mutations in *BIRC3* indicate an aggressive disease course in CLL and MCL. Further, the mutational status of *CXCR4* in lymphoplasmacytic lymphoma (LPL) has been shown to influence disease burden and response to targeted therapies, further underscoring the prognostic role of genomic alterations [11,12].

Given these findings, molecular profiling through NGS is now an essential tool for risk assessment, helping to identify high-risk genomic alterations that may necessitate more aggressive treatment approaches. However, despite its proven utility in prognostic and therapeutic decision making, the role of NGS in diagnosis of mature B-cell neoplasms remains less clearly defined. Cytogenetics and FCM remain the primary diagnostic modalities. However, there is growing interest in whether targeted NGS panels could complement or even enhance the diagnostic process in lymphoid neoplasms. Jajosky et al. evaluated the clinical utility of a custom-designed 31-gene NGS panel in the routine assessment of lymphoid neoplasms [13]. A total of 147 blood, bone marrow, and tissue specimens were analyzed, of which 81% were B-cell neoplasms, with smaller proportions of T-cell (15%) and NK-cell disorders (3%). The study revealed that B-cell lymphomas harbored distinct genetic alterations that could aid in diagnosis, prognosis, and therapy selection. A pathogenic mutation was detected in 64% of all B-cell cases, with certain genetic variants playing a key role in subclassification and clinical decision making [13].

### Objective

Currently, the diagnostic workup of mature B-cell lymphomas remains heavily reliant on cytogenetics and FCM, which provide essential structural and immunophenotypic insights for disease classification. FISH and karyotyping are well-established techniques for detecting hallmark chromosomal rearrangements, while FCM enables rapid identification of clonality and lineage-specific markers. In contrast, molecular genetic testing has so far been primarily applied in prognostic and risk stratification contexts rather than as a frontline diagnostic tool. Despite the increasing availability of NGS in hematopathology, its role in primary diagnosis remains less defined, and its routine implementation in lymphoma classification has been relatively limited. With this study, we aim to evaluate the diagnostic relevance of targeted NGS in mature B-cell lymphomas with potential bone marrow involvement. Specifically, we seek to determine whether NGS-based mutation profiling can provide additional diagnostic clarity, particularly in cases where IHC, cytogenetics, and FCM yield inconclusive results. As a first step, we will present the technical validation of our NGS approach to ensure the reliability and reproducibility of the method. By assessing mutation frequencies and their correlation with established diagnostic markers, we aim to demonstrate the potential diagnostic role of lymphoid NGS beyond its current application in risk assessment and prognosis.

## 2. Materials and Methods

This study serves a dual purpose: (1) to provide validation data for an expanded NGS panel, and (2) to assess its diagnostic performance in a clinical setting. Our custom NGS panel consists of 39 genes (hybrid-capture enrichment). This panel expands upon the Myeloid Solution™ panel by SOPHiA GENETICS (Lausanne, Switzerland), incorporating 14 key genes relevant to B-cell neoplasms, *BCOR*, *BCORL1*, *BIRC3*, *BRAF*, *BTK*, *CXCR4*, *KRAS*, *MYD88*, *NOTCH1*, *NOTCH2*, *NRAS*, *SF3B1*, *TP53*, and *PLCG2*. These genes were selected based on their diagnostic, prognostic, and therapeutic relevance, as recognized in the WHO-HAEM5 and International Consensus Classification (ICC) of mature B-cell neoplasms. Mutations such as *MYD88 L265P* in LPL and *BRAF V600E* in classic HCL are established WHO diagnostic criteria. *NOTCH1* and *NOTCH2* mutations hold significance in CLL, marginal zone lymphoma (MZL), and DLBCL, contributing to disease progression and response to therapy. Mutations in *TP53* serve as a high-risk marker across multiple B-cell malignancies. *BTK* and *PLCG2* mutations are directly linked to ibrutinib resistance in CLL and MCL. *CXCR4* mutations, found in 30–40% of LPL cases, impact bone marrow involvement and response to Bruton’s tyrosine kinase (BTK) inhibitors, while *BIRC3* mutations are associated with relapsed/refractory CLL and MCL. *BCOR*, *BCORL1* and *SF3B1* mutations have been identified as recurrent alterations in splenic B-cell lymphomas and CLL. *KRAS* and *NRAS* mutations, commonly seen in aggressive lymphomas, play a role in proliferative signaling pathways and have been implicated in disease transfor mation and resistance to targeted therapies [1,14].

### 2.1. Analytical Validation

This study aimed to validate an expanded NGS panel and assess its diagnostic performance in a clinical setting. The Myeloid Solution™ panel (SOPHiA GENETICS, Lausanne, Switzerland), originally designed for myeloid malignancies, was customized by incorporating nine additional genes relevant to B-cell neoplasms. This modification necessitated a formal validation of the in-house assay before its clinical implementation. Validation was performed using 40 samples, 20 patient-derived samples with known mutations, 3 quality control (QC) samples, and 17 patient-derived negative controls. Analytical performance was assessed in terms of sensitivity, specificity, precision, and reproducibility. A predefined false-negative rate exceeding 5% or a false-positive rate above 5% would have indicated an unacceptable risk of erroneous classification and necessitated further optimization and extension of the validation. NGS was carried out as follows: Genomic DNA (gDNA) was extracted from EDTA-anticoagulated peripheral blood and bone marrow samples using the EZ1 Advanced XL platform (QIAGEN, Hilden, Germany) with the 350 µL DNA extraction kit, yielding an elution volume of 200 µL. DNA concentration was measured with a Qubit Fluorometer (Thermo Fisher Scientific, Waltham, MA, USA) using the dsDNA HS Assay Kit, while DNA integrity and fragment size distribution were evaluated with the Agilent 4200 TapeStation system (Agilent Technologies, Santa Clara, CA, USA). Library preparation was carried out using a custom-designed targeted sequencing panel (Myeloid Solution™, SOPHiA GENETICS), covering 39 genes relevant to hematologic malignancies. The workflow included enzymatic DNA fragmentation, followed by end-repair, A-tailing, and adapter ligation with dual-index barcodes. The libraries underwent size selection and purification using AMPure XP beads (Beckman Coulter, Brea, CA, USA), after which they were PCR-amplified to enrich target sequences. Library quality and concentration were reassessed using both the Agilent TapeStation (Agilent Technologies, Santa Clara, CA, USA) and Qubit Fluorometer (Thermo Fisher Scientific, Waltham, MA, USA) before sequencing. Target enrichment was achieved through hybridization with gene-specific probes, followed by capture using streptavidin-coated magnetic beads. A second round of PCR amplification was performed to further enrich for target sequences. The final libraries were sequenced on an Illumina MiSeqDX platform using the MiSeq Reagent Kit v3 (600-cycle). A 0.5% spike-in of PhiX control DNA (Illumina, San Diego, CA, USA) was included in each sequencing run to monitor error rates and ensure high-quality sequencing. Post-sequencing, reads were demultiplexed and converted into FASTQ files using the Illumina Local Run Manager. To ensure the reliability of variant detection, sequencing data were analyzed using the SOPHiA DDM^®^ platform, which performs read alignment, variant calling, and annotation against the hg19 reference genome. The pipeline employs proprietary algorithms for variant detection, but key quality metrics were assessed to maintain data integrity.

Variant filtering was based on predefined quality thresholds to exclude low-confidence calls. Variants were retained only if they met the following criteria: minimum coverage of 1000×, a minimum of 10 supporting reads, and a Phred quality score exceeding Q20. The minimum variant allele frequency (VAF) threshold was set at 3%, ensuring robust detection of clinically relevant somatic mutations. Homopolymeric regions exceeding 10 nucleotides were excluded to prevent sequencing artifacts.

The QA report generated by SOPHiA DDM^®^ provided an additional layer of validation by assessing coverage metrics, read mapping efficiency, and sequencing quality. Each sequencing run was evaluated using Horizon reference standards (HD829 and HD734), with internal controls ensuring a detection limit of ≥3% VAF for known mutations. Cluster density, error rates, and Phred scores were reviewed to confirm adherence to sequencing quality benchmarks.

To assess the overall sequencing performance, the Illumina Sequencing Analysis Viewer (SAV) tool was utilized to monitor key parameters, including % ≥ Q30 bases (>70%), cluster density (600–1100 K/mm^2^), cluster passing filter rate (>90%), and total sequencing yield (≥7.0 G). Alignment rates and mapped reads were reviewed to ensure data consistency.

Variant classification followed the American College of Medical Genetics and Genomics (ACMG) guidelines, with annotation performed using ClinVar, gnomAD, COSMIC, and Varsome databases.

### 2.2. Clinical Validation

The diagnostic utility of the expanded NGS panel was evaluated in a cohort of 226 patients suspected of having B-cell lymphomas with potential bone marrow involvement. Patients were recruited retrospectively, details on patient recruitment for the clinical validation are presented in Figure 1. The diagnostic performance of the customized NGS panel was evaluated across five key clinical scenarios. First, its ability to detect WHO-HAEM5-defined molecular markers (diagnostic criteria) was assessed in HCL (*BRAF* mutations), LPL (*MYD88/CXCR4* mutations), and SBLPN (absence of *BRAF* mutations). Second, the utility of NGS in lymphoma exclusion diagnostics was analyzed in patients undergoing staging or presenting with suspected lymphoma but negative immunocytology, identifying cases where NGS provided additional diagnostic clarity. Third, the panel’s performance was examined in immunocytologically or Ig-rearranged confirmed lymphomas, determining the mutation detection rates across different subtypes. Fourth, cases with limited or non-interpretable cytogenetic findings were evaluated to assess whether NGS could compensate for missing cytogenetic data. Lastly, the diagnostic yield of NGS was compared to cytogenetics in cases with ambiguous microscopic findings.

### 2.3. Statistical Analysis

The analytical performance of the NGS panel was assessed through multiple statistical parameters. Sensitivity and specificity were calculated using well-characterized reference samples with previously confirmed genetic variants. Precision was evaluated by analyzing intra-run and inter-run reproducibility, ensuring consistent detection of mutations across different sequencing batches. To assess the concordance between NGS-derived variant allele frequencies (VAF) and those obtained from orthogonal validation methods, correlation analyses were performed using Pearson’s and Spearman’s correlation coefficients, as appropriate. A Bland–Altman plot was generated to visualize agreement between methods and detect potential systematic biases in VAF quantification. For the clinical validation component, descriptive statistics were used to summarize mutation frequencies across lymphoma subtypes. Continuous variables were reported as the mean, standard deviation (SD), and median, while categorical variables were presented as absolute counts and percentages. Fisher’s exact test was performed to evaluate whether the difference in diagnostic yield between NGS and cytogenetics was statistically significant. Moreover, to quantify the likelihood of detecting a relevant mutation using NGS versus cytogenetics, an Odds Ratio (OR) analysis was conducted.

## 3. Results

### 3.1. Analytical Validation

The analytical performance of the customized NGS panel was assessed using 40 validation samples, including 20 patient-derived samples with known mutations, 3 quality control (QC) samples, and 17 mutation-negative controls. The panel achieved 100% sensitivity and specificity, with all 63 expected mutations correctly identified and no false positives detected in mutation-negative samples. Intra-run and inter-run validation confirmed high reproducibility, with identical mutation profiles obtained across independent sequencing runs. Correlation analysis demonstrated a strong agreement between NGS-derived variant allele frequencies (VAF) and reference values (Pearson’s correlation coefficient, r = 0.995). A Bland–Altman analysis showed minimal bias, with 95% limits of agreement ranging from −3.71% to +3.75%. A scatter plot depicting Pearson’s correlation and a Bland–Altman plot is presented in Figure 2.

### 3.2. NGS Performance in WHO-Relevant B-Cell Lymphomas

The ability of NGS to detect WHO-HAEM5-defined lymphoma-associated mutations was assessed in HCL, LPL, and SBLPN. Among the 17 patients with HCL, NGS successfully identified *BRAF* mutations in 100% of cases, confirming its diagnostic reliability. In SBLPN, where the absence of *BRAF* mutations serves as a defining molecular feature, all four cases tested negative, aligning with WHO criteria. For LPL, where *MYD88* and *CXCR4* mutations are recognized as diagnostic markers, at least one pathogenic variant was detected in all 12 patients. Specifically, 10 patients harbored *MYD88* mutations, one had a *CXCR4* mutation, and one exhibited both *MYD88* and *CXCR4* mutations (Figure 3).

### 3.3. NGS Utility in Lymphoma Exclusion Diagnostics

NGS was evaluated in 121 patients undergoing lymphoma staging or presenting with suspected lymphoma but negative immunocytology. Among these cases, five patients were later diagnosed with B-cell lymphoma via cytogenetics, highlighting the limitations of standard immunocytology in ruling out disease. In particular, one patient initially classified as lymphoma negative was found to harbor a *BCORL1* mutation (VAF 24.6%) through NGS. This prompted a re-evaluation of bone marrow aspirates, ultimately leading to a revised diagnosis of marginal zone lymphoma.

### 3.4. NGS Performance in Immunocytologically or Ig-Rearranged Confirmed Lymphomas

To evaluate the broader diagnostic coverage of NGS, mutation detection rates were analyzed in 105 immunocytologically confirmed and/or Ig-rearranged lymphomas. Among these cases, 68 (65%) exhibited at least one mutation, while 37 (35%) had no detectable mutations. The detection rates varied across subtypes, with LPL and HCL exhibiting 100% mutation positivity. CLL exhibited a 62% mutation detection rate with the most common alterations occurring in *BIRC3*, *NOTCH1*, *TP53*, and *SF3B1*.

In MCL, mutations were primarily observed in *TP53*, *KRAS* and *NOTCH1*, while in MZL, *NOTCH2* mutations were most frequently detected. No mutations were identified in *PLCG2* across all analyzed cases and no mutations were detected in follicular lymphoma, as detailed in Table 1 and Table 2.

### 3.5. NGS Performance in Cases with Limited or Non-Interpretable Cytogenetics

A total of 23 patients had suggestive immunocytological findings for B-cell lymphoma but lacked interpretable cytogenetic data due to insufficient metaphases or non-informative interphase nuclei. NGS successfully identified at least one pathogenic variant in 14 of these cases (61%), suggesting that NGS can serve as a valuable alternative in cases where cytogenetics is inconclusive or technically restricted.

### 3.6. NGS Performance in Cases with Limited Morphological Assessment

The diagnostic yield of NGS vs. cytogenetics was compared in 20 cases with ambiguous microscopic findings. NGS identified relevant mutations in 14 of 20 cases (70%), whereas cytogenetics detected abnormalities in only 6 of 20 cases (30%). This difference was statistically significant (*p* = 0.0256), indicating that NGS provided a significantly higher diagnostic yield than cytogenetics alone. Furthermore, an OR of 5.44 was calculated, demonstrating that NGS was 5.44-fold more likely than cytogenetics to detect a relevant genetic abnormality.

## 4. Discussion

The role of NGS in mature B-cell lymphoma hematopathology has primarily centered on prognosis and risk assessment, with its diagnostic potential remaining underexplored. While NGS has enhanced prognostic stratification, its integration as a frontline diagnostic tool is still evolving. Our study aimed to bridge this gap by evaluating the diagnostic performance of an expanded lymphoid NGS panel through both analytical and clinical validation.

Analytical validation confirmed the panel’s robustness and reliability for molecular profiling. Key performance metrics—sensitivity, specificity, reproducibility, and precision—exceeded established thresholds, validating its diagnostic applicability. Testing 40 samples, including patient-derived specimens with known mutations, negative controls, and quality control samples, yielded a 100% concordance rate with reference data. The panel met predefined validation criteria: sensitivity and specificity above 95%, false-negative and false-positive rates below 5%, and stringent sequencing depth and variant allele frequency concordance. Had these criteria not been met, we would have expanded the validation cohort. The sample size of 40 was based on existing recommendations and the limited expansion of an already validated panel, which included only nine additional genes. Given that validation sample sizes in the literature typically range from 20 to 80, our selected cohort was appropriate for assessing this panel’s analytical performance [15]. Moreover, the reproducibility of our findings across independent sequencing runs reinforces the reliability of our approach.

Our study further supports the diagnostic utility of NGS for the classification of WHO-defined B-cell neoplasms, particularly LPL, HCL, and SBLPN [16]. These entities have been increasingly recognized as genetically distinct, with specific molecular markers. *MYD88 L265P* is an established diagnostic marker for LPL, detected in >90% of cases and directly linked to Waldenström’s macroglobulinemia. This mutation drives NF-κB activation, reinforcing its diagnostic significance. Additionally, *CXCR4* mutations, found in 30–40% of LPL cases as subclonal alterations, influence disease presentation, bone marrow involvement, and response to BTK inhibitors [17,18,19]. Our study identified both *MYD88* and *CXCR4* mutations, further validating the molecular characterization of LPL and NGS as a reliable diagnostic tool.

Similarly, *BRAF V600E* mutations are nearly pathognomonic for classic HCL, with studies confirming its presence in almost 100% of cases. The absence of this mutation in HCL-variant and splenic B-cell neoplasms further underscores its diagnostic relevance. Our study successfully identified *BRAF V600E* in all confirmed HCL cases, corroborating its specificity for this entity. Beyond *BRAF*, additional mutations such as *KLF2* and *CDKN1B*, present in a subset of HCL cases, have been suggested to cooperate with BRAF in disease pathogenesis [20,21]. These findings align with our study results, further validating the role of NGS in distinguishing classical HCL from mimicking entities. For SBLPN, the defining molecular criterion remains the absence of *BRAF* mutations, as reaffirmed by WHO-HAEM5. In our study, all tested SBLPN cases were *BRAF* wild-type, aligning with classification standards. While no recurrent mutations were detected using our panel, NGS plays a key role in confirming the absence of HCL-defining alterations, supporting its value in differential diagnosis [22,23].

NGS in hematopathology has primarily been used for prognosis and risk assessment, with its diagnostic potential remaining underexplored.

Our findings underscore NGS as a valuable complement to traditional diagnostics, particularly when microscopy and cytogenetics yield inconclusive results. NGS proved essential in cases with non-interpretable cytogenetic data, where standard karyotyping failed due to insufficient metaphases or non-informative interphase nuclei. The detection of pathogenic variants in a significant proportion of such cases highlights NGS’s utility, especially in settings constrained by technical limitations [10,13].

Furthermore, our results show that NGS outperforms conventional cytogenetics in detecting relevant genetic alterations in cases with ambiguous microscopic findings. The significantly higher diagnostic yield of NGS, evidenced by its ability to identify mutations in a greater proportion of cases (*p* = 0.0256, OR = 5.44), suggests that molecular profiling should be considered when histopathological and immunophenotypic assessments are inconclusive [13]. This observation aligns with previous findings highlighting the limitations of conventional cytogenetics in detecting small structural variants or point mutations, which can be missed due to the resolution constraints of karyotyping and FISH [24]. Zhang et al. demonstrated that molecular genetic methods significantly improve *FLT3* and *NPM1* mutation detection in AML, especially in cases with a normal karyotype where cytogenetics yield no significant findings [25]. Similarly, Kayser and Levis highlight the role of NGS in identifying mutations undetectable by traditional methods, directly impacting risk stratification and treatment decisions [26]. Integrating NGS into routine workflows could further enhance diagnostic accuracy, particularly in low-grade lymphoproliferative disorders, where morphological overlap complicates classification [10].

From a clinical perspective, NGS enhances the detection of genetic markers relevant to B-cell neoplasms, directly impacting diagnosis. When standard immunocytology fails to confirm lymphoma, NGS provides an additional layer of evidence, enabling more precise disease classification. A notable example in our cohort was the identification of a *BCORL1* mutation in a patient initially classified as lymphoma negative, prompting a re-evaluation of bone marrow aspirates and a revised diagnosis of marginal zone lymphoma [27,28].

Despite its benefits, routine NGS implementation faces challenges. Variant interpretation, particularly for variants of unknown significance, remains a key limitation. While our study confirmed the high analytical validity of our customized NGS panel, broader standardization is needed to ensure reproducibility and inter-laboratory concordance across sequencing platforms and bioinformatics pipelines [10,29]. Integrating machine learning and advanced computational tools could refine NGS data interpretation, improving the identification of clinically actionable variants and reducing diagnostic uncertainty [30,31,32].

## 5. Conclusions

The diagnostic landscape of mature B-cell lymphomas is undergoing a paradigm shift with the integration of NGS into routine hematopathology. While traditional methodologies such as cytogenetics, flow cytometry, and histopathology remain indispensable, our findings underscore the increasing relevance of molecular profiling in cases where conventional approaches yield inconclusive results. Our results emphasize the potential of NGS as a complementary tool to enhance accuracy in modern hematopathological diagnostics of mature B-cell lymphomas.

## Figures and Tables

**Figure 1 diagnostics-15-00727-f001:**
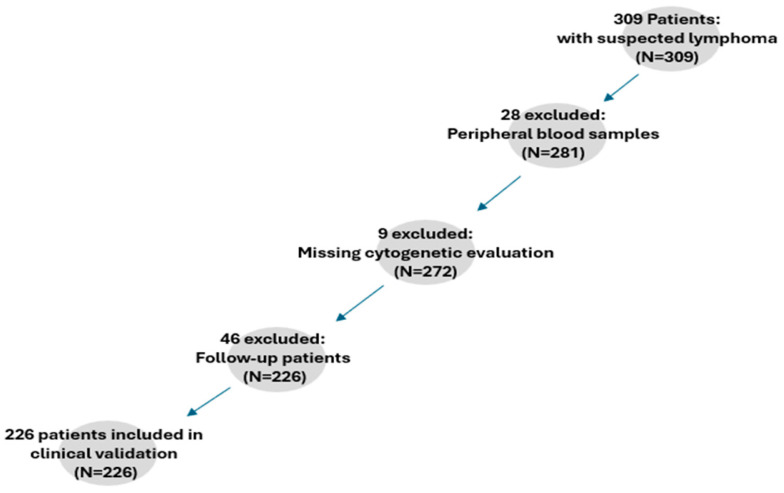
A flowchart of the patient recruitment process of the clinical validation.

**Figure 2 diagnostics-15-00727-f002:**
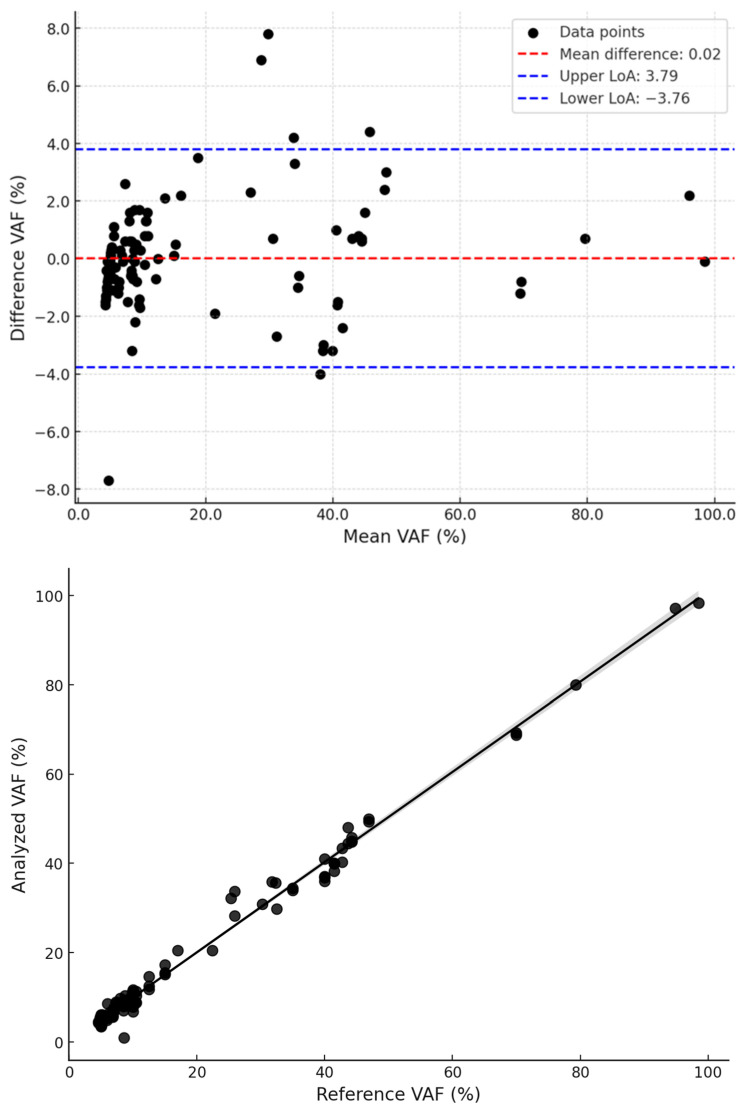
A linear trend in the scatter plot was obtained with data points closely aligning along the regression line, while the Bland–Altman plot confirms the high level of agreement with minimal bias.

**Figure 3 diagnostics-15-00727-f003:**
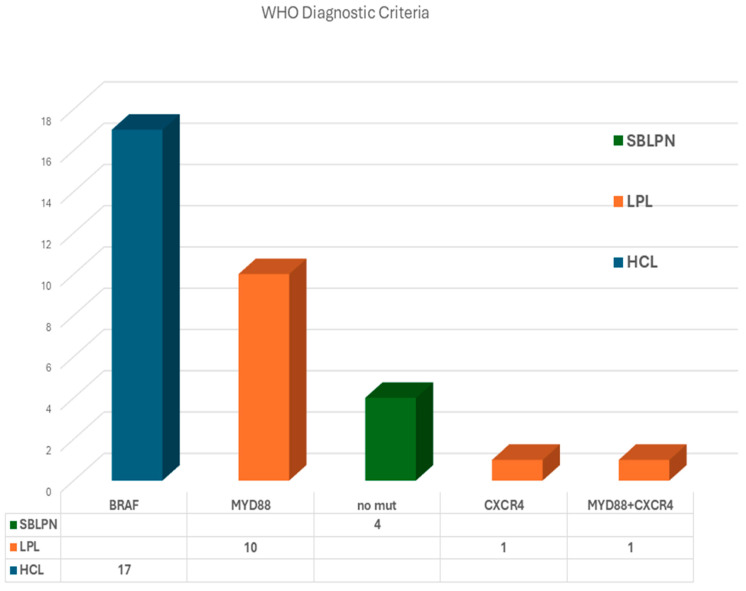
Mutation distribution across lymphoma entities hairy cell leukemia (HCL), splenic B-cell lymphoma with prominent nucleoli (SBLPN), and lymphoplasmacytic lymphoma (LPL). The data confirm the alignment of our customized 39-gene panel with WHO HAEM5 diagnostic criteria, demonstrating its ability to differentiate lymphoma relevant subtypes based on their molecular profiles.

**Table 1 diagnostics-15-00727-t001:** Overview of mutation detection across different lymphoma entities. This table summarizes the proportion of cases harboring at least one detectable genetic alteration in the tested genes. ‘Mutated’ indicates the presence of at least one pathogenic or likely pathogenic variant in any gene included in the panel.

Diagnosis	Non-Mutated	Mutated	Total	% Non-Mutated	% Mutated
**CLL**	11	18	29	37.93	62.07
**FL**	11	0	11	100.00	0.00
**HCL**	0	17	17	0.00	100.00
**HGBCL**	3	3	6	50.00	50.00
**LPL**	0	12	12	0.00	100.00
**MCL**	6	4	10	60.00	40.00
**MZL**	5	6	11	45.45	54.55
**SBLPN**	4	0	4	100.00	0.00
**B-NHL NOS**	3	2	5	60.00	40.00

**Table 2 diagnostics-15-00727-t002:** The heatmap visualizes the mutation distribution across lymphoma subtypes, highlighting entity-specific genetic alterations.

	BCOR	BIRC3	BRAF	BTK	CXCR4	KRAS	MYD88	NOTCH1	NOTCH2	NRAS	SF3B1	TP53	PLCG2
**B-NOS**	0	0	0	0	1	0	0	1	0	0	0	1	0
**CLL**	2	6	1	1	0	0	0	5	0	1	7	6	0
**HCL**	0	0	17	0	0	0	0	0	0	0	0	0	0
**HGBCL**	1	0	0	0	0	0	0	0	0	0	0	2	0
**LPL**	0	0	0	0	2	0	11	0	0	0	0	0	0
**MCL**	0	0	0	0	0	1	0	1	0	0	0	3	0
**MZL**	0	0	0	0	0	0	0	1	3	0	0	3	0
			0	1	2	3	4	6	8	12	17		

## Data Availability

The original contributions presented in this study are included in this article. Further inquiries can be directed to the corresponding author.

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
