# Peer review of "Diagnostic Implications of NGS-Based Molecular Profiling in Mature B-Cell Lymphomas with Potential Bone Marrow Involvement"

_diagnostics, 2025, doi:10.3390/diagnostics15060727_

Round 1
Reviewer 1 Report
Comments and Suggestions for Authors
The idea of ​​the article is interesting and, in my opinion, makes a certain contribution to expanding the range of diagnostic methods necessary for correct verification of the disease and, as a result, choosing the most optimal method of treatment. As my own experience as a practicing hematologist shows, complex cases with ambiguous interpretation of morphological, immunohistochemical and standard molecular genetic findings are not as few as it might seem at first glance. Therefore, this article is of undoubted practical interest. Moreover, in my opinion, the article fully fulfills the task formulated by the authors. I hope that this work will be continued and, eventually, results will be obtained that will finally secure for the NGS method not only the role of a prognostic, but also a diagnostic tool. The article can be recommended for publication. But ... there is (in my opinion) one irritating factor - a drawn-out, not laconic, discussion, when reading which the thought "when will it finally end" flashes in your head!
Author Response
We sincerely appreciate your valuable feedback on our manuscript. We are particularly grateful for your recognition of the study’s practical relevance and its contribution to expanding diagnostic methods in complex hematological cases. Your acknowledgment that our work successfully fulfills its intended objective and holds promise for further development is highly encouraging.
Most notably, we greatly appreciate your conclusion that "The article can be recommended for publication."
With regard to your comments on the discussion section, we have carefully considered your feedback and have revised this part of the manuscript to improve its conciseness and clarity. In this updated version, we have refined the text to be more succinct while ensuring that the key arguments and interpretations from the first draft remain intact.
Reviewer 2 Report
Comments and Suggestions for Authors
In this article, Strasser et al. used the NGS-Based Molecular Profiling for hematological malignancies to perform diagnostics. They found NGS is helpful for diagnostics and reduce bone marrow procedures. The manuscript is well written. I have no further commnets.
Author Response
Reply: We sincerely appreciate your positive evaluation of our manuscript and your recognition of its clarity and contribution to the field.
Reviewer 3 Report
Comments and Suggestions for Authors
Overall, the authors want to show the diagnostic utility of a custom gene panel in lymphoma. The authors could provide the details of the assay as in number of genes, primers per gene if they used nested one or not. Additional, justification for the genes chosen should be given as drug target, frequently mutated etc. What does Sophia genetic pipeline employ regarding variant annotation. Is there any information regarding mapping, variant calling or is it proprietary? Could the authors provide numbers for the line 154 ? “Identified variants were filtered based on read depth, allele frequency, and quality metrics, ensuring robust data interpretation” . It will be useful to to see the filtering criteria values. What does the correlation matrix below table 1 signify? Also, the table 1 should be clarified. Does mutated means mutations in any gene, two genes or all genes tested, for example.
Author Response
Reviewer 3: Overall, the authors want to show the diagnostic utility of a custom gene panel in lymphoma. The authors could provide the details of the assay as in number of genes, primers per gene if they used nested one or not. Additional, justification for the genes chosen should be given as drug target, frequently mutated etc.
Reply: Genes were chosen based on their frequent mutation in B-cell malignancies and their importance in WHO and ICC-defined classification schemes. The assay utilizes hybrid-capture enrichment rather than nested PCR, ensuring high specificity and sensitivity for somatic variant detection. This approach allows for a comprehensive analysis of relevant genetic regions without the amplification bias associated with PCR-based methods. (lines 120-138)
Reviewer 3: What does Sophia genetic pipeline employ regarding variant annotation. Is there any information regarding mapping, variant calling or is it proprietary? Could the authors provide numbers for the line 154 ? “Identified variants were filtered based on read depth, allele frequency, and quality metrics, ensuring robust data interpretation” . It will be useful to to see the filtering criteria values.
Reply: we have expanded the manuscript in lines 170–191 to provide additional details on the Sophia Genetics pipeline, including its approach to read alignment, variant calling, and annotation, as well as the key quality metrics reported. We have also specified the variant filtering criteria applied in our analysis, including thresholds for coverage, allele frequency, and read quality to ensure robust data interpretation.
Reviewer 3: What does the correlation matrix below table 1 signify? Also, the table 1 should be clarified. Does mutated means mutations in any gene, two genes or all genes tested, for example.
Reply: The table presents an overview of mutation frequencies across different lymphoma entities, summarizing the proportion of cases harboring at least one detectable genetic alteration. "Mutated" in this context refers to the presence of at least one pathogenic or likely pathogenic variant in any of the genes included in our panel. We have now explicitly stated this in the manuscript for clarity. Additionally, the figure accompanying Table 1 is not a correlation matrix but a simplified heatmap, visually representing the distribution of detected mutations across lymphoma subtypes. We have revised the figure caption and the corresponding text in the manuscript to prevent any misunderstanding.